# COMPRESSIVE RECOVERY DEFENSE: A DEFENSE FRAMEWORK FOR $\ell_0, \ell_2$, AND $\ell_\infty$ NORM ATTACKS.

## 1 ABSTRACT

We provide recovery guarantees for compressible signals that have been corrupted with noise and extend the framework introduced in Bafna et al. (2018) to defend neural networks against $\ell_0$, $\ell_2$, and $\ell_\infty$-norm attacks. In the case of $\ell_0$-norm noise, we provide recovery guarantees for Iterative Hard Thresholding (IHT) and Basis Pursuit (BP). For $\ell_2$-norm bounded noise, we provide recovery guarantees for BP, and for the case of $\ell_\infty$-norm bounded noise, we provide recovery guarantees for a modified version of Dantzig Selector (DS). These guarantees theoretically bolster the defense framework introduced in Bafna et al. (2018) for defending neural networks against adversarial inputs. Finally, we experimentally demonstrate the effectiveness of this defense framework against an array of $\ell_0$, $\ell_2$ and $\ell_\infty$-norm attacks.

## 2 INTRODUCTION

Signal measurements are often corrupted by noise. The theory of compressive sensing (Candes et al. (2006)) allows us to retrieve the original signal from a corrupted measurement, under some structural assumptions on the measurement mechanism and the signal. Let us consider the class of machine learning problems where the inputs are compressible (i.e., approximately sparse) in some domain. For instance, images and audio signals are known to be compressible in their frequency domain and machine learning algorithms have been shown to perform exceedingly well on classification tasks that take such signals as input (Krizhevsky et al. (2012); Sutskever et al. (2014)). However, it was found in Szegedy et al. (2013) that neural networks can be easily forced into making incorrect predictions by adding adversarial perturbations to their inputs; see also Szegedy et al. (2014); Goodfellow et al. (2015); Papernot et al. (2016); Carlini & Wagner (2017). Further, the adversarial perturbations that led to incorrect predictions were shown to be very small (in either $\ell_0$, $\ell_2$, or $\ell_\infty$-norm) and often imperceptible to human beings. For this class of machine learning tasks, we show how to approximately recover original inputs from adversarial inputs and thus defend the neural network $\ell_0$-norm, $\ell_2$-norm and $\ell_\infty$-norm attacks.

In the case of $\ell_0$-norm attacks on neural networks, the adversary can perturb a bounded number of coordinates in the input vector but has no restriction on how much each coordinate is perturbed in absolute value. In the case of $\ell_2$-norm attacks, the adversary can perturb as many coordinates of the input vector as they choose as long as the $\ell_2$-norm of the perturbation vector is bounded. Finally, in $\ell_\infty$-norm attacks, the adversary is only constrained by the amount of noise added to each coordinate of the input vector.

The contribution and structure of this paper is as follows. In Section 3.1, we describe the Compressive Recovery Defense (CRD) framework, a compressive-sensing-based framework for defending neural networks against adversarial inputs. This is essentially the same framework introduced in Bafna et al. (2018), though Bafna et al. (2018) considered only $\ell_0$ attacks. In Section 3.2, we present the recovery algorithms which are used in the CRD framework to approximately recover original inputs from adversarial inputs. These algorithms include standard Basis Pursuit (BP), $(k, t)$-sparse Iterative Hard Thresholding (IHT) and Dantzig Selector (DS) with an additional constraint. In Section 3.3, we state recovery guarantees for the recovery algorithms in the presence of noise bounded in either $\ell_0$, $\ell_2$, or $\ell_\infty$-norm. The guarantees apply to arbitrary $\ell_0$, $\ell_2$, and $\ell_\infty$-norm attacks; they do not require prior knowledge of the adversary's attack strategy. The recovery guarantees are proved rigorously in Appendix A. In Section 4, we experimentally demonstrate the performance of

the CRD framework in defending neural network classifiers on CIFAR-10, MNIST, and Fashion-MNIST datasets against state-of-the-art $\ell_0$, $\ell_2$ and $\ell_\infty$-norm attacks.

**Notation.** Let $x$ be a vector in $\mathbb{C}^N$. Let $S \subseteq \{1, \ldots, N\}$ and $\overline{S} = \{1, \ldots, N\} \setminus S$. The cardinality of $S$ is $|S|$. If $A \in \mathbb{C}^{m \times N}$ is a matrix, then $A_S \in \mathbb{C}^{m \times |S|}$ is the column submatrix of $A$ consisting of the columns indexed by $S$. We denote by $x_S$ either the sub-vector in $\mathbb{C}^S$ consisting of the entries indexed by $S$ or the vector in $\mathbb{C}^N$ that is formed by starting with $x$ and setting the entries indexed by $\overline{S}$ to zero. For example, if $x = [4, 5, -9, 1]^T$ and $S = \{1, 3\}$, then $x_S$ is either $[4, -9]^T$ or $[4, 0, -9, 0]^T$. It will always be clear from context which meaning is intended. Note that, under the second meaning, $x_{\overline{S}} = x - x_S$. The support of $x$, denoted by $\mathrm{supp}(x)$, is the set of indices of the non-zero entries of $x$, i.e., $\mathrm{supp}(x) = \{i \in \{1, \ldots, N\} : x_i \neq 0\}$. The $\ell_0$-quasinorm of $x$, denoted $\|x\|_0$, is defined to be the number of non-zero entries of $x$, i.e. $\|x\|_0 = \mathrm{card}(\mathrm{supp}(x))$. We say that $x$ is $k$-sparse if $\|x\|_0 \leq k$. We use $x_{h(k)}$ to denote a $k$-sparse vector in $\mathbb{C}^N$ consisting of the $k$ largest (in absolute value) entries of $x$ with all other entries zero. For example, if $x = [4, 5, -9, 1]^T$ then $x_{h(2)} = [0, 5, -9, 0]^T$. Note that $x_{h(k)}$ may not be uniquely defined. In contexts where a unique meaning for $x_{h(k)}$ is needed, we can choose $x_{h(k)}$ out of all possible candidates according to a predefined rule (such as the lexicographic order). We also define $x_{t(k)} = x - x_{h(k)}$. If $x = [x_1, x_2]^T \in \mathbb{C}^{2n}$ with $x_1, x_2 \in \mathbb{C}^n$, and if $x_1$ is $k$-sparse and $x_2$ is $t$-sparse, then $x$ is called $(k, t)$-sparse. We define $x_{h(k,t)} = [(x_1)_{h(k)}, (x_2)_{h(t)}]^T$, which is a $(k, t)$-sparse vector in $\mathbb{C}^{2n}$.

## 3 Theory

### 3.1 Compressive Recovery Defense (CRD)

Bafna et al. (2018) introduced a framework for defending machine learning classifiers against $\ell_0$-attacks. We extend the framework to $\ell_2$ and $\ell_\infty$ attacks. The defense framework is based on the theory of compressive sensing, so we call it Compressive Recovery Defense (CRD).

We explain the idea behind the CRD framework in the context of an image classifier. Suppose $x \in \mathbb{C}^n$ is a (flattened) image vector we wish to classify. But suppose an adversary perturbs $x$ with a noise vector $e \in \mathbb{C}^n$. We observe $y = x + e$, while $x$ and $e$ are unknown to us. Let $F \in \mathbb{C}^{n \times n}$ be the Discrete Fourier Transform (DFT) matrix. The Fourier coefficients of $x$ are $\hat{x} = Fx$. It is well-known that natural images are approximately sparse in the frequency domain. So we expect that $\hat{x}$ is approximately sparse, meaning that $\hat{x}_{t(k)}$ is small for some small $k$. We can write

$$y = F^{-1}\hat{x} + e \tag{1}$$

If $\|e\|_2 \leq \eta$ or $\|e\|_\infty \leq \eta$, with $\eta$ small (as in a $\ell_2$ or $\ell_\infty$-attack), then we can use an appropriate sparse recovery algorithm with $y$ and $F^{-1}$ as input to compute a good approximation $x^\#$ to $\hat{x}$. Precise error bounds are given in Section 3.3. Then, since $F$ is unitary, $F^{-1}x^\#$ will be a good approximation (i.e., reconstruction) of $x = F^{-1}\hat{x}$. So we can feed $F^{-1}x^\#$ into the classifier and expect to get the same classification as we would have for $x$. For an $\ell_0$-attack where $e$ is $t$-sparse, the approach is only slightly different. We set $A = [F^{-1}, I]$ and write

$$y = F^{-1}\hat{x} + e = F^{-1}\hat{x}_{h(k)} + e + F^{-1}\hat{x}_{t(k)} = A[\hat{x}_{h(k)}, e]^T + F^{-1}\hat{x}_{t(k)}, \tag{2}$$

so that $[\hat{x}_{h(k)}, e]^T$ is $(k, t)$-sparse. This structure lets us use a sparse recovery algorithm to compute a good approximation to $\hat{x}$, as before. Note that the same idea can be applied with audio signals or other types of data instead of images. Moreover, the DFT can be replaced by any unitary transformation $F$ for which $\hat{x} = Fx$ is approximately sparse. For example, $F$ may be the Cosine Transform, Sine Transform, Hadamard Transform, or another wavelet transform.

We now describe the training and testing procedure for CRD. For each training image $x$, we compute $\hat{x}_{h(k)} = (Fx)_{h(k)}$, and then compute the compressed the image $x' = F^{-1}\hat{x}_{h(k)}$. We then add both $x$ and $x'$ to the training set and train the network in the usual way. Given a (potentially adversarial) test image $y$, we first use a sparse recovery algorithm to compute an approximation $x^\#$ to $\hat{x}$, then we compute the reconstructed image $y' = F^{-1}x^\#$ and feed it into the network for classification.

### 3.2 Recovery Algorithms

We provide the recovery algorithms used in this section. For $\ell_0$-attacks, we set $A = [F^{-1}, I]$ as in (2). Against $\ell_2$ or $\ell_\infty$-attacks, we take $A = F^{-1}$ as in (1).

**Algorithm 1**: $(k, t)$-Sparse Iterative Hard Thresholding (IHT)
Procedure: $IHT(y, A, k, t, T)$
Input: $y \in \mathbb{C}^n$, $A \in \mathbb{C}^{n \times 2n}$, and positive integers $k, t, T$.
$x^{[0]} = 0$
for $i := 0$ to $T$ do
$$x^{[i+1]} = \left(x^{[i]} + A^*(y - Ax^{[i]})\right)_{h(k,t)}$$
return $x^{\#} = x^{[T+1]}$

The IHT algorithm above is used to defend against $\ell_0$-norm attacks. For such attacks, according to (2), the vector we need to recover is $(k, t)$-sparse. Thus this IHT is adapted to the structure of our problem as it uses the thresholding operation $h_{(k,t)}$ that produces $(k, t)$-sparse vectors. This structured IHT was first considered in Baraniuk et al. (2010). It gives better theoretical guarantees and practical performance in our CRD application than the standard IHT, which would instead use the thresholding operation $h_{(k+t)}$ that produces $(k + t)$-sparse vectors. For $\ell_2$ or $\ell_\infty$ attacks, the recovery error for IHT would (in general) be larger due to the need to include a term for the $\ell_2$ norm of the tail of the noise vector $e$. This, in turn, produces worse expected performance of the recovery defense. Therefore we only use Algorithm 1 for $\ell_0$-norm attacks. We note that the results of Theorem 1 allow for values of $k$ and $t$ greater than or equal to Theorem 2.2. of Bafna et al. (2018).

**Algorithm 2**: Basis Pursuit (BP).
Procedure: $BP(y, A, \eta)$.
Input: $y \in \mathbb{C}^m$, $A \in \mathbb{C}^{m \times N}$, and $\eta \geq 0$.
$x^{\#} = \arg\min_{z \in \mathbb{C}^N} \|z\|_1$ subject to $\|Az - y\|_2 \leq \eta$
return $x^{\#}$

We utilize BP for $\ell_0$ and $\ell_2$ norm attacks. In the $\ell_0$ norm case, BP allows us to provide recovery guarantees for larger values of $k$ and $t$ than IHT. For instance, in the case of MNIST and Fashion-MNIST, IHT (equation (4) of Theorem 1) allows us to set $k = 4$ and $t = 3$, whereas BP (equation (7) of Theorem 2) allows us to set $k = 8$ and $t = 8$.

In the case of $\ell_2$ norm attacks, BP is applied with $A = F^{-1}$, a unitary matrix. As unitary matrices are isometries in $\ell_2$ norm, BP provides good recovery guarantees for such matrices, and hence against $\ell_2$ norm attacks.

**Algorithm 3**: Modified Dantzig Selector (DS).
Procedure: $DS(y, A, \eta)$.
Input: $y \in \mathbb{C}^m$, $A \in \mathbb{C}^{m \times N}$, and $\eta \geq 0$.
$x^{\#} = \arg\min_{z \in \mathbb{C}^N} \|z\|_1$ subject to $\|A^*(Az - y)\|_\infty \leq \sqrt{n}\eta$, $\|Az - y\|_\infty \leq \eta$
return $x^{\#}$

We utilize DS for $\ell_\infty$ norm attacks. The standard Dantzig Selector algorithm does not have the additional constraint $\|Az - y\|_\infty \leq \eta$. Our modified Dantzig Selector includes this constraint for the following reason. In our application, $A = F^{-1}$ and we want the reconstruction $Ax^{\#} = F^{-1}x^{\#}$ to be close to the original image $x$, so that they are classified identically. Thus, we want to the search space for $x^{\#}$ to be restricted to those $z \in \mathbb{C}^N$ such that $\|Az - x\|_\infty$ is small. Note, for any $z \in \mathbb{C}^N$, $\|Az - x\|_\infty \leq \|Az - y\|_\infty + \|x - y\|_\infty$. In an $\ell_\infty$-attack, $\|x - y\|_\infty = \|e\|_\infty$ is already small. Thus it suffices to require $\|Az - y\|_\infty$ is small. We experimentally illustrate the improvement in reconstruction due to the additional constraint in Section 4.3 (Figure 4, Table 4).

**Remarks on Reverse-Engineered Attacks.** As observed in Bafna et al. (2018), $x^{[0]}$ in Algorithm 1, can be initialized randomly to defend against a reverse-engineered attack. In the case of Algorithm 2 and Algorithm 3, the minimization problems can be posed as semi-definite programming problems. If solved with interior point methods, one can use random initialization of the central path parameter and add randomness to the stopping criterion. This makes recovery non-deterministic and consequently non-trivial to create a reverse-engineered attack.

## 3.3 RECOVERY GUARANTEES

Let $F \in \mathbb{C}^{n \times n}$ be a unitary matrix and $I \in \mathbb{C}^{n \times n}$ be the identity matrix. Define $A = [F, I] \in \mathbb{C}^{n \times 2n}$ and let $y = A[\hat{x}, e]^T = F\hat{x} + e$, where $\hat{x}, e \in \mathbb{C}^n$. Let $1 \leq k, t \leq n$ be integers.

**Theorem 1** ($\ell_0$-norm IHT). *Assume* $|F_{ij}|^2 \leq \frac{c}{n}$ *and* $e$ *is* $t$-sparse. *Let* $x^{[T+1]} = IHT(y, A, k, t, T)$ *where* $x^{[T+1]} = \left[\hat{x}^{[T+1]}, e^{[T+1]}\right]^T \in \mathbb{C}^{2n}$ *with* $\hat{x}^{[T+1]}, e^{[T+1]} \in \mathbb{C}^n$.

*Define* $\rho := \sqrt{27}\sqrt{\frac{ckt}{n}}, \quad \tau(1-\rho) := \sqrt{3}\sqrt{1 + 2\sqrt{\frac{ckt}{n}}}$. *If* $0 < \rho < 1$, *then:*

$$\|\hat{x}^{[T+1]} - \hat{x}_{h(k)}\|_2 \leq \rho^{(T+1)}\sqrt{\|\hat{x}_{h(k)}\|_2^2 + \|e\|_2^2} + \tau\|\hat{x}_{t(k)}\|_2 \tag{3}$$

*Moreover for any* $0 < \epsilon < 1$ *and any* $T \geq \left(\frac{\log(1/\epsilon) + \log(\sqrt{\|\hat{x}_{h(k)}\|_2^2 + \|e\|_2^2})}{\log(1/\rho)}\right)$, *we get:*

$$\|\hat{x}^{[T+1]} - \hat{x}_{h(k)}\|_2 \leq \tau\|\hat{x}_{t(k)}\|_2 + \epsilon \tag{4}$$

*Now define* $\rho := 2\sqrt{2}\sqrt{\frac{ckt}{n}}, \quad \tau(1-\rho) := 2$. *If* $0 < \rho < 1$, *then:*

$$\|\hat{x}^{[T+1]} - \hat{x}_{h(k)}\|_2 \leq \rho^{(T+1)}\|\hat{x}_{h(k)}\|_2 + \tau(\|\hat{x}_{t(k)}\|_2 + \|e\|_2) \tag{5}$$

*Moreover for any* $0 < \epsilon < 1$ *and any* $T \geq \left(\frac{\log(1/\epsilon) + \log(\|\hat{x}_{h(k)}\|_2)}{\log(1/\rho)}\right)$, *we get:*

$$\|\hat{x}^{[T+1]} - \hat{x}_{h(k)}\|_2 \leq \tau(\|\hat{x}_{t(k)}\|_2 + \|e\|_2) + \epsilon \tag{6}$$

Let us explain how to interpret the recovery guarantees provided by Theorem 1. The inequalities (3), (4), (5), (6) provide an upper bound on the size of $\|\hat{x}^{[T+1]} - \hat{x}_{h(k)}\|_2$. Since $F$ is a unitary matrix, $\|\hat{x}^{[T+1]} - \hat{x}_{h(k)}\|_2$ equals $\|F\hat{x}^{[T+1]} - F\hat{x}_{h(k)}\|_2$, which is the difference between the reconstructed image $F\hat{x}^{[T+1]}$ and the compressed image $F\hat{x}_{h(k)}$ (which is a compressed version of the original image $x$). So the inequalities of Theorem 1 tell us how close the reconstructed image must be to the compressed image, and thus indicates how confident we should be that the classification of the reconstructed image will agree with the classification of the compressed image. In other words, the inequalities tell us how likely it is that the CRD scheme using IHT will be able to recover the correct class of the original image, and thus defend the classifier from the adversarial attack. The presence of the norm of the tail $\hat{x}_{t(k)}$ in the upper bounds indicates that the CRD scheme should be more effective when the original image is closer to being perfectly $k$-sparse in the transformed basis. The ratio $kt/n$ in the upper bounds (via $\rho$ and $\tau$) suggests that smaller values of $k$ and $t$ relative to $n$ (i.e., sparser transformed images $\hat{x}$ and error vectors $e$) will lead to the CRD being more effective. The experiments in Section 4 will demonstrate these phenomena.

Let us compare Theorem 1 to the similar Theorem 2.2 of Bafna et al. (2018). We observe that (3) and (4) allow larger values of $k$ and $t$ than Theorem 2.2 of Bafna et al. (2018). This is because the authors of Bafna et al. (2018) prove their results using Theorem 4 of Baraniuk et al. (2010), which is more restrictive for the values of $k$ and $t$. We do not use Theorem 4 of Baraniuk et al. (2010). Instead we use (a modified form of) Theorem 6.18 of Foucart & Rauhut (2017) to get (3) and (4). Both Theorem 4 of Baraniuk et al. (2010) (used by Bafna et al. (2018)) and Theorem 6.18 of Foucart & Rauhut (2017) (used by us here) take as input the Restricted Isometry Property (RIP) stated in Theorem 7. We and Bafna et al. (2018) both essentially prove the same RIP, although the proof methods are different. We use a standard Gershgorin disc theorem argument to bound eigenvalues, while Bafna et al. (2018) perform a direct estimation using the triangle inequality and AM-GM inequality.

We turn now to (5) and (6), which provide recovery guarantees for larger values of $k$ and $t$ than (3) and (4), at the expense of the extra error term $\|e\|_2$. Our proof of (5) and (6) is novel. It relies on explicitly expanding one iteration of IHT in matrix form and using the structure of the resulting matrix form to bound the approximation error at iteration $T$ in terms of the error at iteration $T - 2$. We then use an inductive argument as in Theorem 6.18 of Foucart & Rauhut (2017) to get (5) and (6).

Next, we consider the recovery error for $\ell_0$-norm bounded noise with BP instead of IHT. We note that since Algorithm 2 is not adapted to the $(k, t)$-sparse structure of vector to be recovered, we do not expect the guarantees to be particularly strong. However, providing bounds for BP is useful as there are cases when BP provides recovery guarantees for when recovering a larger number of coefficients ($k$) and a larger $\ell_0$ noise budget ($t$) than IHT.

**Theorem 2** ($\ell_0$-norm BP). *Assume $|F_{ij}|^2 \leq \frac{c}{n}$. Define*

$$\delta_{k,t} = \sqrt{\frac{ckt}{n}}, \quad \beta = \sqrt{\frac{\max\{k,t\}c}{n}}, \quad \theta = \frac{\sqrt{k+t}}{(1-\delta_{k,t})}\beta, \quad \tau = \frac{\sqrt{1+\delta_{k,t}}}{1-\delta_{k,t}}$$

*If $0 < \delta_{k,t} < 1$ and $0 < \theta < 1$, then for $x^\# = BP(y, A, \|\hat{x}_{t(k)}\|_2)$, we have the error bound*

$$\|\hat{x}^\# - \hat{x}_{h(k)}\|_2 \leq \left(\frac{2\tau\sqrt{k+t}}{1-\theta}\left(1 + \frac{\beta}{1-\delta_{k,t}}\right) + 2\tau\right)\|\hat{x}_{t(k)}\|_2 \quad (7)$$

*where we write $x^\# = [\hat{x}^\#, e^\#]^T \in \mathbb{C}^{2n}$ with $\hat{x}^\#, e^\# \in \mathbb{C}^n$.*

Note that the recovery error in (7) is $O((\sqrt{k+t})\|\hat{x}_{t(k)}\|_2)$, which means that we should not expect recovery to be close when the attacker has a large $\ell_0$ noise budget or when $\hat{x}$ is not sparse. Also observe that the recovered vector $\hat{x}^\#$ is not necessarily $k$-sparse. The recovery error still captures the difference in the original image $F\hat{x}$ and the reconstructed image $F\hat{x}^\#$, where a smaller recovery error should once again indicate that our classifier would make the correct prediction. Our third result covers the case when the noise is bounded in $\ell_2$-norm.

**Theorem 3** ($\ell_2$-norm BP). *If $\|e\|_2 \leq \eta$, then for $x^\# = BP(y, F, \eta)$, we have the error bound*

$$\|x^\# - \hat{x}\|_1 \leq 2\left(\|\hat{x}_{t(k)}\|_1 + 2\sqrt{k}\eta\right) \quad (8)$$

$$\|x^\# - \hat{x}\|_2 \leq \frac{2}{\sqrt{k}}\|\hat{x}_{t(k)}\|_1 + 6\eta \quad (9)$$

Finally, we provide recovery guarantees when the noise is bounded in $\ell_\infty$-norm.

**Theorem 4** ($\ell_\infty$-norm DS). *If $\|e\|_\infty \leq \eta$, then for $x^\# = DS(y, F, \eta)$, we have the error bound*

$$\|x^\# - \hat{x}\|_1 \leq 2\left(\|\hat{x}_{t(k)}\|_1 + 2k\sqrt{n}\eta\right) \quad (10)$$

$$\|x^\# - \hat{x}\|_2 \leq \frac{2}{\sqrt{k}}\|\hat{x}_{t(k)}\|_1 + 6\sqrt{kn}\eta \quad (11)$$

The proofs of Theorem 3 and Theorem 4 are based on standard arguments in compressive sensing that rely on establishing the so-called robust null space property of the matrix. Note that the results of Theorem 3 and Theorem 4 also bound the norm difference of the original image $F\hat{x}$ and the reconstructed image $F\hat{x}^\#$, where $\hat{x}^\#$ has no sparsity guarantees. Next, observe that the results of Theorem 4 incur a factor of $\sqrt{n}$ in the error bounds due to the constraint $\|A^*(Az - y)\|_\infty \leq \sqrt{n}\eta$ in Algorithm 3 which is required to prove the robust null space property. Finally, we note that the additional constraint added to Algorithm 3 does not affect the proof of Theorem 4.

### 3.4 RELATED WORK

The authors of Bafna et al. (2018) introduced the CRD framework which inspired this work. In fact, Theorem 2.2 of Bafna et al. (2018) also provides an approximation error bound for recovery via IHT. Note that a hypothesis $t = O(n/k)$ has accidentally been dropped from their Theorem 2.2, though it appears in their Lemma 3.6. By making the implied constants explicit in the argument of Bafna et al. (2018), one sees that their Theorem 2.2 is essentially the same as (3) and (4) in Theorem 1 above. For more details, see the proof of Theorem 1 in Appendix A. Note that our recovery error bounds for IHT in (5) and (6) of Theorem 1 do not have analogs in Bafna et al. (2018). They hold for larger values of $k$ and $t$ at the expense of the additional error term $\|e\|_2$.

Other works that provide guarantees include (Hein & Andriushchenko (2017)) and (Cisse et al. (2017)) where the authors frame the problem as one of regularizing the Lipschitz constant of a network and give a lower bound on the norm of the perturbation required to change the classifier decision. The authors of Sinha et al. (2017) use robust optimization to perturb the training data and provide a training procedure that updates parameters based on worst case perturbations. A similar approach to (Sinha et al. (2017)) is (Wong & Kolter (2017)) in which the authors use robust optimization to provide lower bounds on the norm of adversarial perturbations on the training data.

In Lecuyer et al. (2018), the authors use techniques from Differential Privacy (Dwork et al. (2014))in order to augment the training procedure of the classifier to improve robustness to adversarial inputs. Another approach using randomization is Li et al. (2018) in which the authors add i.i.d. Gaussian noise to the input and provide guarantees of maintaining classifier predictions as long as the $\ell_2$-norm of the attack vector is bounded by a function that depends on the output of the classifier.

Most defenses against adversarial inputs do not come with theoretical guarantees. Instead, a large body of research has focused on finding practical ways to improve robustness to adversarial inputs by either augmenting the training data (Goodfellow et al. (2015)), using adversarial inputs from various networks (Tramèr et al. (2017)), or by reducing the dimensionality of the input (Xu et al. (2017)). For instance, Madry et al. (2017) use robust optimization to make the network robust to worst case adversarial perturbations on the training data. However, the effectiveness of their approach is determined by the amount and quality of training data available and its similarity to the distribution of the test data. An approach similar to ours but without any theoretical guarantees is (Samangouei et al. (2018)). In this work, the authors use Generative Adversarial Networks (GANs) to estimate the distribution of the training data and during inference, use a GAN to reconstruct a non-adversarial input that is most similar to a given test input. We now provide a brief overview on the field of compressive sensing.

Though some component ideas originated earlier in other fields, the field of compressive sensing was initiated with the work of Candès et al. (2006) and Donoho et al. (2006) in which the authors studied the problem of reconstructing sparse signals using only a small number of measurements with the choice of a random matrix. The reconstruction was performed using $\ell_1$-minimization (i.e., Basis Pursuit) which was shown to produce sparse solutions even in presence of noise; see also Donoho & Elad (2003; 2006); Donoho & Huo (2001). Some of the earlier work in extending compressive sensing to perform stable recovery with deterministic matrices was done by Candes & Tao (2005) and Candes et al. (2006), where the authors showed that recovery of sparse vectors could be performed as long as the measurement matrix satisfied a restricted isometry hypothesis. Blumensath & Davies (2009) introduced IHT as an algorithm to recover sparse signals which was later modified in Baraniuk et al. (2010) to reduce the search space as long as the sparsity was structured. The standard DS algorithm was introduced in Candes et al. (2007) in order to perform stable recovery in the presence of $\ell_\infty$ noise.

## 4 EXPERIMENTS

All of our experiments are conducted on CIFAR-10 (Krizhevsky (2009)), MNIST (LeCun), and Fashion-MNIST (Xiao et al. (2017)) datasets with pixel values of each image normalized to lie in $[0, 1]$. Each experiment is conducted on a set of 1000 points sampled uniformly at random from the test set of the respective dataset. For every experiment, we use the Discrete Cosine Transform (DCT) and the Inverse Discrete Cosine Transform (IDCT) denoted by the matrices $F \in \mathbb{R}^{n \times n}$ and $F^T \in \mathbb{R}^{n \times n}$ respectively. That is, for an adversarial image $y \in \mathbb{R}^{\sqrt{n} \times \sqrt{n}}$, such that, $y = x + e$, we let $\hat{x} = Fx$, and $x = F^T \hat{x}$, where $x, \hat{x} \in \mathbb{R}^n$ and $e \in \mathbb{R}^n$ is the noise vector. For an adversarial image $y \in \mathbb{R}^{\sqrt{n} \times \sqrt{n} \times c}$, that contains $c$ channels, we perform recovery on each channel independently by considering $y_m = x_m + e_m$, where $\hat{x}_m = F x_m, x_m = F^T \hat{x}_m$ for $m = 1, \ldots, c$. The value $k$ denotes the number of largest (in absolute value) DCT coefficients used for reconstruction of each channel, and the value $t$ denotes the $\ell_0$ noise budget for each channel. We implement Algorithm 2 and Algorithm 3 using the open source library CVXPY (Diamond & Boyd (2016)).

We now outline the neural network architectures used for experiments in Section 4.1 and 4.2. For CIFAR-10, we use the network architecture of He et al. (2016) while the network architecture for MNIST and Fashion-MNIST datasets is provided in Table 5 of the Appendix. We train our networks using the Adam optimizer for CIFAR-10 and the AdaDelta optimizer for MNIST and Fashion-MNIST. In both cases, we use a cross-entropy loss function. We train the each neural network according to the CRD framework stated in Section 3.1. The code to reproduce our experiments is available here: `https://github.com/anonymousiclrcompressive/iclr2020`.

| Orig. Acc. | OPA. Acc | IHT. Acc. |
|:---:|:---:|:---:|
| 77.4% | 0.0% | 71.8% |

Table 1: Effectiveness of CRD against OPA. The first column lists the accuracy of the network on original images and the OPA Acc. column shows the network's accuracy on adversarial images. The IHT. Acc. column shows the accuracy of the network on images reconstructed using IHT.

## 4.1 DEFENSE AGAINST $\ell_0$-NORM ATTACKS

This section is organized as follows: first we examine CRD against the One Pixel Attack (OPA) (Su et al. (2019)) for CIFAR-10. We only test the attack on CIFAR-10 as it is most effective against natural images and does not work well on MNIST or FASHION-MNIST. We note that this attack satisfies the theoretical constraints for $t$ provided in Theorem 1, hence allowing us to test how well CRD works within existing guarantees. Once we establish the effectiveness of CRD against OPA, we then test it against two other $\ell_0$-norm bounded attacks: Carlini and Wagner (CW) $\ell_0$-norm attack (Carlini & Wagner (2017)) and the Jacobian based Saliency Map Attack (JSMA) (Papernot et al. (2016)).

### 4.1.1 ONE PIXEL ATTACK

We first resize all CIFAR-10 images to $125 \times 125 \times 3$ while maintaining aspect ratios to ensure that the data falls under the hypotheses of Theorem 1 even for large values of $k$. The OPA attack perturbs exactly one pixel of the image, leading to an $\ell_0$ noise budget of $t = 3$ per image. The $\ell_0$ noise budget of $t = 3$ per image allows us to use $k = 275$ per channel. Table 1 shows that OPA is very effective against natural images and forces the network to mis-classify all previously correctly classified inputs.



Original        OPA        IHT-Rec

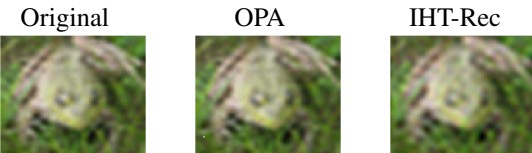



Figure 1: The original image is shown in the first column, the adversarial image in the second column, and image reconstructed using IHT is shown in the third column.

We test the performance of CRD in two ways: a) reconstruction quality b) network performance on reconstructed images. In order to analyse the reconstruction quality of Algorithm 1, we do the following: for each test image, we use OPA to perturb the image and then use Algorithm 1 to approximate its largest (in absolute value) $k = 275$ DCT co-efficients. We then perform the IDCT on these recovered co-efficients to generate reconstructed images. We illustrate reconstruction on a randomly selected image from the test set in Figure 1.

Noting that Algorithm 1 leads to high quality reconstruction, we now test whether network accuracy improves on these reconstructed images. To do so, we feed these reconstructed images as input to the network and report its accuracy in Table 1. We note that network performance does indeed improve as network accuracy goes from $0.0\%$ to $71.8\%$ using Algorithm 1. Therefore, we conclude that CRD provides a substantial improvement in accuracy in against OPA.

### 4.1.2 CW-$\ell_0$ ATTACK AND JSMA

Having established the effectiveness of CRD against OPA, we move onto the CW $\ell_0$-norm attack and JSMA. We note that even when $t$ is much larger than the hypotheses of Theorem 1 and Theorem 2, we find that Algorithms 1 and 2 are still able to defend the network. We hypothesize that this maybe related to the behavior of the RIP of a matrix for "most" vectors as opposed to the RIP for all vectors, and leave a more rigorous analysis for a follow up work.

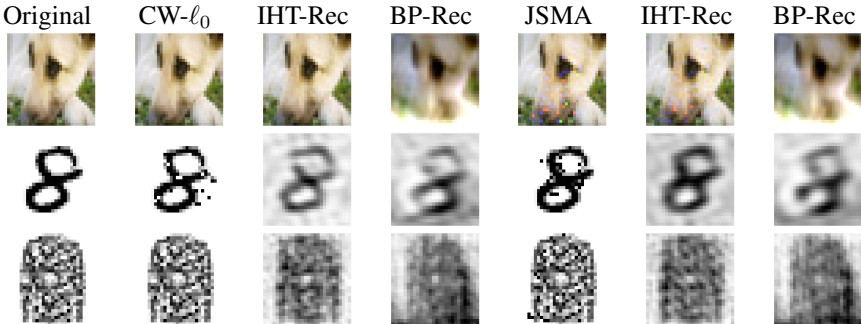

Figure 2: Reconstruction quality of images using IHT and BP. The first column shows randomly selected original images from the test set, while the second and fifth column show the adversarial images. Reconstructions using IHT are labeled IHT-Rec and using BP are labeled BP-Rec. We show reconstructions in columns three, four, six, and seven.

| Dataset | Orig. | C&W $\ell_0$ | | | | JSMA | | | |
|---------|-------|--------------|--|--|--|------|--|--|--|
| | Acc. | $t_{avg}$ | Acc. | IHT Acc. | BP Acc. | $t_{avg}$ | Acc. | IHT Acc. | BP Acc. |
| CIFAR-10 | 84.9% | 18 | 8.7% | 83.0% | 67.0% | 34 | 2.7% | 63.2% | 67.3% |
| MNIST | 98.8% | 15 | 0.9% | 84.2% | 55.9% | 17 | 56.5 % | 90.1% | 67.4% |
| F-MNIST | 91.8% | 16 | 5.27% | 84.1% | 71.4% | 17 | 62.6 % | 83.3% | 72.0% |

Table 2: The $t_{avg}$ column lists the average adversarial budget for each attack. The Orig. Acc column lists the accuracy of the network on original test inputs, the Acc. columns under C&W $\ell_0$ and JSMA list network accuracy on adversarial inputs. The IHT Acc. and the BP Acc. columns list the accuracy of the network on inputs that have been corrected using IHT and BP respectively.

We follow the procedure described in Section 4.1.1 to analyze the quality of reconstructions for Algorithm 1 and Algorithm 2 in Fig 2. In each case it can be seen that both algorithms provide high quality reconstructions for values of $t$ that are well outside the hypotheses required by Theorem 1 and Theorem 2. We report these $t$ values and the improvement in network performance on reconstructed adversarial images using CRD in Table 2.

## 4.2 Defense against $\ell_2$-norm attacks

In the case of $\ell_2$-norm bounded attacks, we use the CW $\ell_2$-norm attack (Carlini & Wagner (2017)) and the Deepfool attack (Moosavi-Dezfooli et al. (2016)) as they have been shown to be the most powerful. We note that Theorem 3 does not impose any restrictions on $k$ or $t$ and therefore the guarantees of equations (8) and (9) are applicable for recovery in all experiments of this section.

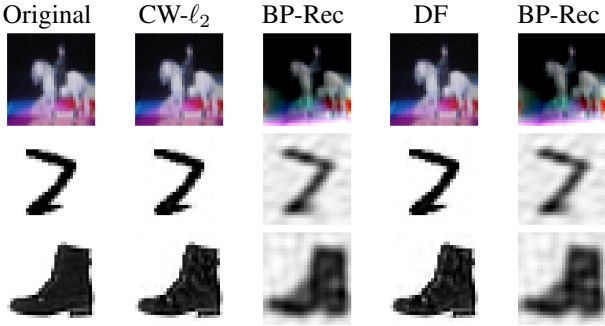

Figure 3: Reconstruction quality of images using BP. The first columns shows the original images, while the adversarial images are shown in the second and fourth column. The reconstructions are shown in columns three and five.

| Dataset | Orig. | C&W $\ell_2$ | | | Deepfool | | |
|---|---|---|---|---|---|---|---|
| | Acc. | $\ell_{2_{avg}}$ | Acc. | BP Acc. | $\ell_{2_{avg}}$ | Acc. | BP Acc. |
| CIFAR-10 | 84.9% | 0.12 | 8.7% | 72.3% | 0.11 | 7.7% | 71.6% |
| MNIST | 99.17% | 1.35 | 0.9% | 92.4% | 1.72 | 1.1 % | 90.7% |
| Fashion-MNIST | 90.3% | 0.61 | 5.4% | 78.3% | 0.63 | 5.5 % | 76.4% |

Table 3: The $\ell_{2_{avg}}$ column lists the average $\ell_2$-norm of the attack vector. The Orig. Acc column lists the accuracy of the network on original test inputs, while the Acc. columns under C&W $\ell_2$ and DF columns report network accuracy on adversarial inputs. BP Acc. columns lists the accuracy of the network on inputs reconstructed using BP.

The reconstruction quality is shown in Figure 3. It can be noted that reconstruction using Algorithm 2 is of high quality for all three datasets. In order to check whether this high quality reconstruction also leads to improved performance in network accuracy, we test each network on reconstructed images using Algorithm 2. We report the results in Table 3 and note that Algorithm 2 provides a substantial improvement in network accuracy for each dataset and each attack method used.

## 4.3 DEFENSE AGAINST $\ell_\infty$-NORM ATTACKS

For $\ell_\infty$-norm bounded attacks, we use the BIM attack (Kurakin et al. (2016)) as it is has been shown to be very effective and also allows us to control the $\ell_\infty$-norm of the attack vector explicitly. We note that while the CW $\ell_\infty$-norm attack (Carlini & Wagner (2017)) has the ability to create attack vectors with $\ell_\infty$-norm less than or equal to BIM, it is computationally expensive and also does not allow one to pre-specify a value for the $\ell_\infty$-norm of an attack vector. Therefore, we limit our experimental analysis to the BIM attack. Note that for any attack vector $e$, $\|e\|_2 \leq \sqrt{n}\|e\|_\infty$ hence allowing $\ell_\infty$-norm attacks to create attack vectors with large $\ell_2$-norm. Therefore, we could expect reconstruction quality and network accuracy to be lower when compared to $\ell_2$-norm attacks.

Original     With Constraint     No Constraint

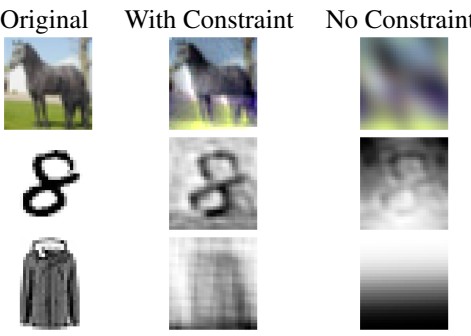

Figure 4: Comparison of images reconstructed using Algorithm 3 (With Constraint) with images reconstructed using DS without the additional constraint (No Constraint).

In figure 4, we compare the reconstruction quality of images reconstructed with Algorithm 3 to those reconstructed using DS without the additional constraint. As can be noted from the figure, images reconstructed using DS without the additional constraint may not produce meaningful images. This is also reflected in Table 4, which shows that the accuracy of the network is roughly random on images reconstructed without the additional constraint.

We show examples of original images, adversarial images, and their reconstructions using Algorithm 3 in Figure 5. Finally, we report the network performance on reconstructed inputs using Algorithm 3 in Table 4 and also compare this to the performance on inputs reconstructed using DS without the additional constraint. We note that Algorithm 3 provides an increase in network performance against reconstructed adversarial inputs. However, the improvement in performance is not as substantial as it was against $\ell_0$ or $\ell_2$-norm attacks.

| Dataset | Orig. Acc. | BIM | | | |
|---------|------------|-----|------|------------------|---------|
| | | $\ell_{\infty_{avg}}$ | Acc. | Modified DS Acc. | DS Acc. |
| CIFAR-10 | 84.9% | 0.015 | 7.4% | 49.4% | 17.6% |
| MNIST | 99.17% | 0.15 | 4.9% | 74.7% | 10% |
| Fashion-MNIST | 90.3% | 0.15 | 5.3% | 57.5% | 11.1% |

Table 4: The $\ell_{\infty_{avg}}$ column lists the $\ell_\infty$-norm of each attack vector, Orig. Acc. and BIM Acc. columns list the accuracy of the network on the original and adversarial inputs respectively, and the Modified DS Acc. column lists the accuracy of the network on inputs reconstructed using Algorithm 3. We also show accuracy of the network on images reconstructed with DS (without the additional constraint) in the DS Acc. column.

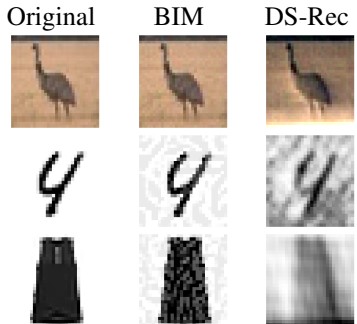

Figure 5: Reconstruction quality of images using DS. The first column shows the original images, while the second columns shows adversarial images and the third columns shows reconstructions using Algorithm 3 respectively.

## 5 CONCLUSION

We provided recovery guarantees for corrupted signals in the case of $\ell_0$-norm, $\ell_2$-norm, and $\ell_\infty$-norm bounded noise. We were able to utilize these results in CRD and improve the performance of neural networks substantially in the case of $\ell_0$-norm, $\ell_2$-norm and $\ell_\infty$-norm bounded noise. While $\ell_0$-norm attacks don't always satisfy the constraints required by Theorem 1 and Theorem 2, we showed that CRD is still able to provide a good defense for values of $t$ much larger than allowed in the guarantees. The guarantees of Theorem 3 and Theorem 4 were applicable in all experiments and CRD was shown to improve network performance for all attacks.

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

## A  APPENDIX

### A.1  RESTRICTED ISOMETRY PROPERTY

We first establish the restricted isometry property for certain structured matrices. First, we give some definitions.

**Definition 5.** Let $A$ be a matrix in $\mathbb{C}^{m \times N}$, let $M \subseteq \mathbb{C}^N$, and let $\delta \geq 0$. We say that $A$ satisfies the $M$-restricted isometry property (or M-RIP) with constant $\delta$ if

$$(1 - \delta)\|x\|_2^2 \leq \|Ax\|_2^2 \leq (1 + \delta)\|x\|_2^2$$

for all $x \in M$.

**Definition 6.** We define $M_k$ to be the set of all $k$-sparse vectors in $\mathbb{C}^N$ and similarly define $M_{k,t}$ to be the set of $(k, t)$-sparse vectors in $\mathbb{C}^{2n}$. In other words, $M_{k,t}$ is the following subset of $\mathbb{C}^{2n}$:

$$M_{k,t} = \left\{ x = [x_1 \ x_2]^T \in \mathbb{C}^{2n} : x_1 \in \mathbb{C}^n, x_2 \in \mathbb{C}^n, \|x_1\|_0 \leq k, \|x_2\|_0 \leq t \right\}$$

We define $S_{k,t}$ to be the following collection of subsets of $\{1, \ldots, 2n\}$:

$$S_{k,t} = \left\{ S_1 \cup S_2 : S_1 \subseteq \{1, \ldots, n\}, S_2 \subseteq \{n+1, \ldots, 2n\}, \mathrm{card}(S_1) \leq k, \mathrm{card}(S_2) \leq t \right\}$$

Note that $S_{k,t}$ is the collection of supports of vectors in $M_{k,t}$.

**Theorem 7.** Let $A = [F \ I] \in \mathbb{C}^{n \times 2n}$, where $F \in \mathbb{C}^{n \times n}$ is a unitary matrix with $|F_{ij}|^2 \leq \frac{c}{n}$ and $I \in \mathbb{C}^{n \times n}$ is the identity matrix. Then

$$\left(1 - \sqrt{\frac{ckt}{n}}\right) \|x\|_2^2 \leq \|Ax\|_2^2 \leq \left(1 + \sqrt{\frac{ckt}{n}}\right) \|x\|_2^2 \tag{12}$$

for all $x \in M_{k,t}$. In other words, $A$ satisfies the $M_{k,t}$-RIP property with constant $\sqrt{\frac{ckt}{n}}$.

*Proof.* In this proof, if $B$ denotes an matrix in $\mathbb{C}^{n \times n}$, then $\lambda_1(B), \ldots, \lambda_n(B)$ denote the eigenvalues of $B$ ordered so that $|\lambda_1(B)| \leq \cdots \leq |\lambda_n(B)|$. It suffices to fix an $S = S_1 \cup S_2 \in S_{k,t}$ and prove (12) for all non-zero $x \in \mathbb{C}^S$.

Since $A_S^* A_S$ is normal, there is an orthonormal basis of eigenvectors $u_1, \ldots, u_{k+t}$ for $A_S^* A_S$, where $u_i$ corresponds to the eigenvalue $\lambda_i(A_S^* A_S)$. For any non-zero $x \in \mathbb{C}^S$, we have $x = \sum_{i=1}^{k+t} c_i u_i$ for some $c_i \in \mathbb{C}$, so

$$\frac{\|Ax\|_2^2}{\|x\|_2^2} = \frac{\langle A_S^* A_S x, x \rangle}{\langle x, x \rangle} = \frac{\sum_{i=1}^{k+t} \lambda_i(A_S^* A_S) c_i^2}{\sum_{i=1}^{k+t} c_i^2}. \tag{13}$$

Thus it will suffice to prove that $|\lambda_i(A_S^* A_S) - 1| \leq \sqrt{\frac{ckt}{n}}$ for all $i$. Moreover,

$$|\lambda_i(A_S^* A_S) - 1| = |\lambda_i(A_S^* A_S - I)| = \sqrt{\lambda_i\left((A_S^* A_S - I)^* (A_S^* A_S - I)\right)} \tag{14}$$

where the last equality holds because $A_S^* A_S - I$ is normal. By combining (13) and (14), we see that (12) will hold upon showing that the eigenvalues of $(A_S^* A_S - I)^* (A_S^* A_S - I)$ are bounded by $ckt/n$.

So far we have not used the structure of $A$, but now we must. Observe that $(A_S^* A_S - I)^* (A_S^* A_S - I)$ is a block diagonal matrix with two diagonal blocks of the form $X^* X$ and $X X^*$. Therefore the three matrices $(A_S^* A_S - I)^* (A_S^* A_S - I)$, $X^* X$, and $X X^*$ have the same non-zero eigenvalues. Moreover, $X$ is simply the matrix $F_{S_1}$ with those rows not indexed by $S_2$ deleted. The hypotheses on $F$ imply that the entries of $X^* X$ satisfy $|(X^* X)_{ij}| \leq \frac{ct}{n}$. So the Gershgorin disc theorem implies that each eigenvalue $\lambda$ of $X^* X$ and (hence) of $(A_S^* A_S - I)^* (A_S^* A_S - I)$ satisfies $|\lambda| \leq \frac{ckt}{n}$.

□

A.2 ITERATIVE HARD THRESHOLDING

First we present Theorem 8 and then use it to prove Theorem 1.

**Theorem 8.** *Let $A \in \mathbb{C}^{n \times 2n}$ be a matrix. Let $1 \leq k, t \leq n$ be positive integers and suppose $\delta_3$ is a $M_{3k,3t}$-RIP constant for $A$ and that $\delta_2$ is a $M_{2k,2t}$-RIP constant for $A$. Let $x \in \mathbb{C}^{2n}$, $r \in \mathbb{C}^n$, $y = Ax + r$, and $S \in S_{k,t}$. Letting $x^{[T+1]} = IHT(y, A, k, t, T)$, if $\delta_3 < 1/\sqrt{3}$, then we have the approximation error bound*

$$\|x^{[T+1]} - x_S\|_2 \leq \rho^{(T+1)}\|x^{[0]} - x_S\|_2 + \tau\|Ax_{\overline{S}} + r\|_2$$

*where $\rho := \sqrt{3}\delta_3 < 1$ and $(1-\rho)\tau = \sqrt{3}\sqrt{1+\delta_2} \leq 2.18$. Thus, the first term on the right goes to $0$ as $T$ goes to $\infty$.*

Theorem 8 is a modification of Theorem 6.18 of Foucart & Rauhut (2017). More specifically, Theorem 6.18 of Foucart & Rauhut (2017) considers $M_{3k}$, $M_{2k}$, and $S_k$ in place of $M_{3k,3t}$ and $M_{2k,2t}$ and $S_{k,t}$ and any dimension $N$ in place of $2n$. The proofs are very similar, so we omit the proof of Theorem 8. We will now prove a lemma that will be required for the proof of Theorem 1. For the proof of Lemma 9 and Theorem 1, we use the following convention: let $A \in \mathbb{C}^{m \times N}$ be a matrix, then, we denote by $(A)_S$, the $m \times N$ matrix that is obtained by starting with $A$ and zeroing out the columns indexed by $\overline{S}$. Note that $(A)_S = A - (A)_{\overline{S}}$.

**Lemma 9.** *Let $F \in \mathbb{C}^{n \times n}$ be a unitary matrix with $|F_{ij}|^2 \leq \frac{c}{n}$ and let $S \subseteq [n]$ be a index set with $|S| = t$. Then for any $k$-sparse vector $z \in \mathbb{C}^n$, we have:*

$$\|(F^*)_S F z\|_2^2 \leq \frac{ktc}{n}\|z\|_2^2$$

*Proof of Lemma 9.* First note that $(F^*)_S \in \mathbb{C}^{n \times n}$ contains only $t$ non-zero columns since $|S| = t$ Therefore, we have $|((F^*)_S F)_{ij}| \leq \frac{tc}{n}$ since $|F_{ij}|^2 \leq \frac{c}{n}$. Further, since the non-zero columns of $(F^*)_S$ are orthogonal to each other, we get $((F^*)_S)^*(F^*)_S = (I)_S$, where $I \in \mathbb{C}^{n \times n}$ is the identity matrix. Using this, we have for any $w \in \mathbb{C}^n$,

$$\|(F^*)_S F w\|_2^2 = \langle (F^*)_S F w, (F^*)_S F w \rangle = \langle ((F^*)_S F)^* (F^*)_S F w, w \rangle = \langle (F^*)_S F w, w \rangle = |\langle (F^*)_S F w, w \rangle|$$

Now let $V \subseteq [n]$ be any index set with cardinality $k$, that is $|V| = k$ and let $z \in \mathbb{C}^n$ be any vector supported on $V$. We then get,

$$\|(F^*)_S F z\|_2^2 = |\langle (F^*)_S F z, z \rangle| = \left| \sum_{k \in V} z_k^* \left( \sum_{j \in V} ((F^*)_S F)_{kj} z_j \right) \right| \leq \sum_{k \in V} |z_k^*| \left( \sum_{j \in V} |((F^*)_S F)_{kj}| |z_j| \right)$$

$$\leq \sum_{k \in V} |z_k^*| \left( \frac{tc}{n} \sum_{j \in V} |z_j| \right)$$

$$= \frac{tc}{n}\|z\|_1^2 \leq \frac{ktc}{n}\|z\|_2^2$$

where we use the fact that $z$ is $k$-sparse for the last inequality. □

Now we provide the proof for Theorem 1.

*Proof of Theorem 1.* Theorem 7 implies that the statement of Theorem 8 holds with

$\delta_3 = \sqrt{\frac{c \cdot 3k \cdot 3t}{n}}$ and $\delta_2 = \sqrt{\frac{c \cdot 2k \cdot 2t}{n}}$. Noting that $y = A[\hat{x}_{h(k)} \ e]^T + F\hat{x}_{t(k)}$, where $[\hat{x}_{h(k)} \ e]^T \in M_{k,t}$, set $x^{[T+1]} = IHT(y, A, k, t, T)$ and apply Theorem 8 with $x = [\hat{x}_{h(k)} \ e]^T$, $r = F\hat{x}_{t(k)}$, and $S = \text{supp}(x)$. Letting $x^{[T+1]} = [\hat{x}^{[T+1]} \ e^{[T+1]}]^T$, use the facts that $\|\hat{x}^{[T+1]} - \hat{x}_{h(k)}\|_2 \leq \|x^{[T+1]} - x_S\|_2$ and $\|F\hat{x}_{t(k)}\|_2 = \|\hat{x}_{t(k)}\|_2$. That will give (3). Letting $T = \left( \frac{\log(1/\epsilon) + \log(\sqrt{\|\hat{x}_{h(k)}\|_2^2 + \|e\|_2^2})}{\log(1/\rho)} \right)$,

gives $\rho^T \sqrt{\|\hat{x}_{h(k)}\|_2^2 + \|e\|_2^2} \leq \epsilon$, which can be substituted in (3) to get (4). Noting that $\|e^{[T]} - e\|_2 \leq \tau \|\hat{x}_{t(k)}\|_2 + \epsilon$, we can use the same reasoning as used in Bafna et al. (2018) to get:

$$\|\hat{x}^{[T+1]} - \hat{x}_{h(k)}\|_\infty \leq \sqrt{\frac{2ct}{n}} \left(\tau \|\hat{x}_{t(k)}\|_2 + \epsilon\right) \qquad (15)$$

$$\|\hat{x}^{[T+1]} - \hat{x}_{h(k)}\|_2 \leq \sqrt{\frac{4ckt}{n}} \left(\tau \|\hat{x}_{t(k)}\|_2 + \epsilon\right) \qquad (16)$$

which are the essentially the same as the results of Theorem 2.2 in Bafna et al. (2018).

Now we prove (5). Write $x^{[T]} = (z^{[T]})_{h(k,t)}$, where $z^{[T]} = x^{[T-1]} + A^*(y - Ax^{[T-1]})$. Further, write $z^{[T]} = [z_1^{[T]} \ z_2^{[T]}]^T \in \mathbb{C}^{2n}$, where $z_1^{[T]}, z_2^{[T]} \in \mathbb{C}^n$. Note that $\hat{x}^{[T]} = (z_1^{[T]})_{h(k)}$. Therefore, we have $z_1^{[T]} = F^*(y - e^{[T-1]})$, where $e^{[T-1]} = (y - F\hat{x}^{[T-2]})_{h(t)}$. Now let $S$ be the set of indices selected by the hard thresholding operation $h(t)$ to get $e^{[T-1]}$. Then observe that $z_1^{[T]} = F^*(y - (y - F\hat{x}^{[T-2]})_S)$. Next, note that $\|z_1^{[T]} - \hat{x}^{[T]}\|_2^2 \leq \|z_1^{[T]} - \hat{x}_{h(k)}\|_2^2$ as $\hat{x}^{[T]}$ is a best $k$-sparse approximation to $z_1^{[T]}$. We can thus write,

$$\|(z_1^{[T]} - \hat{x}_{h(k)}) - (\hat{x}^{[T]} - \hat{x}_{h(k)})\|_2^2 = \|z_1^{[T]} - \hat{x}_{h(k)}\|_2^2 - 2\mathrm{Re}\langle z_1^{[T]} - \hat{x}_{h(k)}, \hat{x}^{[T]} - \hat{x}_{h(k)}\rangle + \|\hat{x}^{[T]} - \hat{x}_{h(k)}\|_2^2$$

Therefore, we have,

$$\|\hat{x}^{[T]} - \hat{x}_{h(k)}\|_2^2 \leq 2\mathrm{Re}\langle z_1^{[T]} - \hat{x}_{h(k)}, \hat{x}^{[T]} - \hat{x}_{h(k)}\rangle$$
$$\leq 2|\langle z_1^{[T]} - \hat{x}_{h(k)}, \hat{x}^{[T]} - \hat{x}_{h(k)}\rangle|$$
$$\leq 2\|z_1^{[T]} - \hat{x}_{h(k)}\|_2 \|\hat{x}^{[T]} - \hat{x}_{h(k)}\|_2$$

If $\|\hat{x}^{[T]} - \hat{x}_{h(k)}\|_2 > 0$, then $\|\hat{x}^{[T]} - \hat{x}_{h(k)}\|_2 \leq 2\|z_1^{[T]} - \hat{x}_{h(k)}\|_2$. Now note that

$$z_1^{[T]} = \hat{x} + F^*e - F^*(F(\hat{x} - \hat{x}^{[T-2]}) + e)_S$$
$$= \hat{x} + F^*e - (F^*)_S(F(\hat{x} - \hat{x}^{[T-2]}) + e)$$
$$= \hat{x} + (F^* - (F^*)_S)e - (F^*)_S F(\hat{x} - \hat{x}^{[T-2]})$$

Using the fact that $(F^*)_{\overline{S}} = F^* - (F^*)_S$, we can simplify the above to get:

$$\|z_1^{[T]} - \hat{x}_{h(k)}\|_2 = \|(F^*)_{\overline{S}} F\hat{x}_{t(k)} + (F^*)_{\overline{S}}e - (F^*)_S F(\hat{x}_{h(k)} - \hat{x}^{[T-2]})\|_2$$

Therefore,

$$\|\hat{x}^{[T]} - \hat{x}_{h(k)}\|_2 \leq 2\left(\|(F^*)_{\overline{S}} F\|_{2\to2}\|\hat{x}_{t(k)}\|_2 + \|(F^*)_{\overline{S}}\|_{2\to2}\|e\|_2 + \|(F^*)_S F(\hat{x}_{h(k)} - \hat{x}^{[T-2]})\|_2\right)$$
$$\leq 2\left(\|\hat{x}_{t(k)}\|_2 + \|e\|_2\right) + 2\|(F^*)_S F(\hat{x}_{h(k)} - \hat{x}^{[T-2]})\|_2$$

where we use $\|(F^*)_{\overline{S}}\|_{2\to2} \leq \|F^*\|_{2\to2} = 1$. Now since $\hat{x}_{h(k)} - \hat{x}^{[T-2]}$ is $2k$-sparse, we can use the result of Lemma 9 to get:

$$\|\hat{x}^{[T]} - \hat{x}_{h(k)}\|_2 \leq 2\left(\|\hat{x}_{t(k)}\|_2 + \|e\|_2\right) + 2\left(\sqrt{\frac{2ktc}{n}}\right)\|\hat{x}^{[T-2]} - \hat{x}_{h(k)}\|_2$$

Now let $\rho = 2\sqrt{2}\sqrt{\frac{ktc}{n}}, \tau(1 - \rho) = 2$ and note that if $\rho < 1$, we can use induction on $T$ to get (5). Then for any $0 < \epsilon < 1$ and any $T \geq \left(\frac{\log(1/\epsilon) + \log(\|\hat{x}_{h(k)}\|_2)}{\log(1/\rho)}\right)$, we have $\rho^T(\|\hat{x}_{h(k)}\|_2) \leq \epsilon$ which gives us (6).

$\square$

### A.3 BASIS PURSUIT

**Definition 10.** The matrix $A \in \mathbb{C}^{m \times N}$ satisfies the robust null space property with constants $0 < \rho < 1, \tau > 0$ and norm $\|\cdot\|$ if for every set $S \subseteq [N]$ with $\text{card}(S) \leq s$ and for every $v \in \mathbb{C}^N$ we have

$$\|v_S\|_1 \leq \rho\|v_{\overline{S}}\|_1 + \tau\|Av\|$$

**Definition 11.** The matrix $A \in C^{m \times N}$ satisfies the $\ell_q$ robust null space property of order $s$ with constants $0 < \rho < 1, \tau > 0$ and norm $\|\cdot\|$ if for every set $S \subseteq [N]$ with $\text{card}(S) \leq s$ and for every $v \in \mathbb{C}^N$ we have

$$\|v_S\|_q \leq \frac{1}{s^{1-1/q}}\rho\|v_{\overline{S}}\|_1 + \tau\|Av\|$$

Note that if $q = 1$ then this is simply the robust null space property.

The proof of Theorem 2 requires the following theorem (whose full proof is given in the Foucart & Rauhut (2017)).

**Theorem 12** (Theorem 4.33 in Foucart & Rauhut (2017)). *Let $a_1, \ldots, a_N$ be the columns of $A \in \mathbb{C}^{m \times N}$, let $x \in \mathbb{C}^N$ with $s$ largest absolute entries supported on $S$, and let $y = Ax + e$ with $\|e\|_2 \leq \eta$. For $\delta, \beta, \gamma, \theta, \tau \geq 0$ with $\delta < 1$, assume that:*

$$\|A_S^* A_S - I\|_{2\to2} \leq \delta, \quad \max_{l \in \overline{S}} \|A_S^* a_l\|_2 \leq \beta,$$

*and that there exists a vector $u = A^* h \in \mathbb{C}^N$ with $h \in \mathbb{C}^m$ such that*

$$\|u_S - \text{sgn}(x_S)\|_2 \leq \gamma, \quad \|u_{\overline{S}}\|_\infty \leq \theta, \quad \text{and } \|h\|_2 \leq \tau\sqrt{s}.$$

*If $\rho := \theta + \frac{\beta\gamma}{(1-\delta)} < 1$, then a minimizer $x^\#$ of $\|z\|_1$ subject to $\|Az - y\|_2 \leq \eta$ satisfies:*

$$\|x^\# - x\|_2 \leq \frac{2}{(1-\rho)}\left(1 + \frac{\beta}{(1-\delta)}\right)\|x_{\overline{S}}\|_1 + \left(\frac{2(\mu\gamma + \tau\sqrt{s})}{1-\rho}\left(1 + \frac{\beta}{1-\delta}\right) + 2\mu\right)\eta$$

*where $\mu := \frac{\sqrt{1+\delta}}{1-\delta}$ and $\text{sgn}(x)_i = \begin{cases} 0, & x_i = 0 \\ 1, & x_i > 0. \\ -1. & x_i < 0 \end{cases}$*

**Lemma 13.** *Let $A \in \mathbb{C}^{n \times 2n}$, if $\|Ax\|_2^2 \leq (1+\delta)\|x\|_2^2$ for all $x \in M_{k,t}$, then, $\|A_S^* A_S - I\|_{2\to2} \leq \delta$, for any $S \in S_{k,t}$.*

*Proof.* Let $S \in S_{k,t}$ be given. Then for any $x \in \mathbb{C}^S$, we have

$$\|A_S x\|_2^2 - \|x\|_2^2 \leq \delta\|x\|_2^2$$

We can re-write this as : $\|A_S x\|_2^2 - \|x\|_2^2 = \langle A_S x, A_S x \rangle - \langle x, x \rangle = \langle (A_S^* A_S - I)x, x \rangle$. Noting that $A_S^* A_S - I$ is Hermitian, we have:

$$\|A_S^* A_S - I\|_{2\to2} = \max_{x \in \mathbb{C}^S \setminus \{0\}} \frac{\langle (A_S^* A_S - I)x, x \rangle}{\|x\|_2^2} \leq \delta$$

$\square$

***Proof of Theorem 2.*** We will derive (7) by showing that the matrix A satisfies all the hypotheses in Theorem 12 for every vector in $M_{k,t}$.

First note that by Theorem 7, $A$ satisfies the $M_{k,t}$-$RIP$ property with constant $\delta_{k,t} := \sqrt{\frac{ckt}{n}}$. Therefore, by Lemma 13, for any $S \in S_{k,t}$, we have $\|A_S^* A_S - I\|_{2\to2} \leq \delta_{k,t}$. Since $A_S^* A_S$ is a positive semi-definite matrix, it has only non-negative eigenvalues that lie in the range $[1 - \delta_{k,t}, 1 + \delta_{k,t}]$. Since $\delta_{k,t} < 1$ by assumption, $A_S^* A_S$ is injective. Thus, we can set: $h = A_S(A_S^* A_S)^{-1}\text{sgn}(x_S)$ and get:

$$\|h\|_2 = \|A_S(A_S^* A_S)^{-1}\text{sgn}(x_S)\|_2 \leq \|A_S\|_{2\to2}\|(A_S^* A_S)^{-1}\|_{2\to2}\|\text{sgn}(x_S)\|_2 \leq \tau\sqrt{k+t}$$

where $\tau = \frac{\sqrt{1+\delta_{k,t}}}{1-\delta_{k,t}}$ and we have used the following facts: since $\|A_S^* A_S - I\|_{2\to 2} \leq \delta_{k,t} < 1$, we get that $\|(A_S^* A_S)^{-1}\|_{2\to 2} \leq \frac{1}{1-\delta_{k,t}}$ and that the largest singular value of $A_S$ is less than $\sqrt{1+\delta_{k,t}}$. Now let $u = A^* h$, then $\|u_S - \text{sgn}(x_S)\|_2 = 0$. Now we need to bound the value $\|u_{\overline{S}}\|_\infty$. Denoting row $j$ of $A_{\overline{S}}^* A_S$ by the vector $v_j$, we see that it has at most $\max\{k,t\}$ non-zero entries and that $|(v_j)_l|^2 \leq \frac{c}{n}$ for $l = 1, \ldots, (k+t)$. Therefore, for any element $(u_{\overline{S}})_j$, we have:

$$|(u_{\overline{S}})_j| = |\langle (A_S^* A_S)^{-1} \text{sgn}(x_S), (v_j)^* \rangle| \leq \|(A_S^* A_S)^{-1}\|_{2\to 2} \|\text{sgn}(x_S)\|_2 \|v_j\|_2 \leq \frac{\sqrt{k+t}}{1-\delta_{k,t}} \sqrt{\frac{\max\{k,t\}c}{n}}$$

Defining $\beta := \sqrt{\frac{\max\{k,t\}c}{n}}$ and $\theta := \frac{\sqrt{k+t}}{1-\delta_{k,t}}\beta$, we get $\|u_{\overline{S}}\|_\infty \leq \theta < 1$ and also observe that $\max_{l\in\overline{S}} \|A_S^* a_l\|_2 \leq \beta$. Therefore, all the hypotheses of Theorem 12 have been satisfied. Note that $y = F\hat{x} + e = A[\hat{x}_{h(k)} \ e]^T + F\hat{x}_{t(k)}$, Therefore, setting $x^{\#} = \text{BP}(y, A, \|\hat{x}_{t(k)}\|_2)$, we use the fact $\|F\hat{x}_{t(k)}\|_2 = \|\hat{x}_{t(k)}\|_2$ combined with the bound in Theorem 12 to get (7):

$$\|\hat{x}^{\#} - \hat{x}_{h(k)}\|_2 \leq \left( \frac{2\tau\sqrt{k+t}}{1-\theta} \left( 1 + \frac{\beta}{1-\delta_{k,t}} \right) + 2\tau \right) \|\hat{x}_{t(k)}\|_2$$

where we write $x^{\#} = [\hat{x}^{\#}, e^{\#}]^T$ with $\hat{x}^{\#}, e^{\#} \in \mathbb{C}^n$. $\qquad\square$

We now focus on proving Theorem 3. In order to do so, we will need some lemmas that will be used in the main proof.

**Lemma 14.** *If a matrix $A \in \mathbb{C}^{m\times N}$ satisfies the $\ell_2$ robust null space property for $S \subset [N]$, with $card(S) = s$, then it satisfies the $\ell_1$ robust null space property for $S$ with constants $0 < \rho < 1, \tau' := \tau\sqrt{s} > 0$.*

*Proof.* For any $v \in \mathbb{C}^N$, $\|v_S\|_2 \leq \frac{\rho}{\sqrt{s}}\|v_{\bar{S}}\|_1 + \tau\|Av\|$. Then, using the fact that $\|v_S\|_1 \leq \sqrt{s}\|v_S\|_2$, we get: $\|v_S\|_1 \leq \rho\|v_{\bar{S}}\|_1 + \tau\sqrt{s}\|Av\|$.

$\qquad\square$

**Lemma 15** (Theorem 4.20 in Foucart & Rauhut (2017)). *If a matrix $A \in \mathbb{C}^{m\times N}$ satisfies the $\ell_1$ robust null space property (with respect to $\|.\|$) and for $0 < \rho < 1$ and $\tau > 0$ for $S \subset [N]$, then:*

$$\|z - x\|_1 \leq \frac{1+\rho}{1-\rho}(\|z\|_1 - \|x\|_1 + 2\|x_{\bar{S}}\|_1) + \frac{2\tau}{1-\rho}\|A(z-x)\|$$

*for all $z, x \in \mathbb{C}^N$.*

**Lemma 16** (Proposition 2.3 in Foucart & Rauhut (2017)). *For any $p > q > 0$ and $x \in \mathbb{C}^n$,*

$$\inf_{z\in M_k} \|x - z\|_p \leq \frac{1}{(k)^{\frac{1}{q}-\frac{1}{p}}}\|x\|_q$$

***Proof of Theorem 3.*** Let $0 < \rho < 1$ be arbitrary. Since $F$ is a unitary matrix, for any $S \subseteq [n]$ and $v \in \mathbb{C}^n$, we have

$$\|v_S\|_2 \leq \frac{\rho}{\sqrt{k}}\|v_{\overline{S}}\|_1 + \tau\|v\|_2 = \frac{\rho}{\sqrt{k}}\|v_{\overline{S}}\|_1 + \tau\|Fv\|_2 \qquad (17)$$

where $\tau = 1$. Now let $S \subseteq [n]$ such that $\text{card}(S) \leq k$. Then, $F$ satisfies the $\ell_2$ robust null space property for $S$. Next, using Lemma 14 we get $\|v_S\|_1 \leq \rho\|v_{\bar{S}}\|_1 + \tau\sqrt{k}\|Fv\|_2$ for all $v \in \mathbb{C}^n$. Now let $x^{\#} = \text{BP}(y, F, \eta)$, then we know $\|x^{\#}\|_1 \leq \|\hat{x}\|_1$. Fixing $S \subseteq [n]$ to be the support of $\hat{x}_{h(k)}$ and using Lemma 15 , we get:

| Layer | Type | Properties |
|---|---|---|
| 1 | Convolution | 32 channels, $3 \times 3$ Kernel, No padding |
| 2 | Convolution | 64 channels, $3 \times 3$ Kernel, No padding, Dropout with $p = 0.5$ |
| 3 | Max-pooling | $2 \times 2$, Dropout with $p = 0.5$ |
| 4 | Fully Connected | 128 neurons, Dropout with $p = 0.5$ |
| 5 | Fully Connected | 10 neurons |

Table 5: Network architecture used for MNIST and Fashion-MNIST datasets in Section 4.1 and Section 4.2. The first four layers use ReLU activations while the last layer uses a softmax activation.

$$\|x^{\#} - \hat{x}\|_1 \leq \frac{1+\rho}{1-\rho}(\|x^{\#}\|_1 - \|\hat{x}\|_1 + 2\|\hat{x}_{t(k)}\|_1) + \frac{2\tau\sqrt{k}}{1-\rho}\|F(x^{\#} - \hat{x})\|_2$$

$$\leq \frac{1+\rho}{1-\rho}\left(2\|\hat{x}_{t(k)}\|_1\right) + \frac{2\tau\sqrt{k}}{1-\rho}\|F(x^{\#} - \hat{x})\|_2$$

$$\leq \frac{1+\rho}{1-\rho}\left(2\|\hat{x}_{t(k)}\|_1\right) + \frac{4\tau\sqrt{k}}{1-\rho}\|e\|_2$$

$$\leq \frac{1+\rho}{1-\rho}\left(2\|\hat{x}_{t(k)}\|_1\right) + \frac{4\tau\sqrt{k}}{1-\rho}\eta$$

Letting $\rho \to 0$ and recalling that $\tau = 1$ gives (8). Now let $S$ be the support of $(x^{\#} - \hat{x})_{h(k)}$. Note $\|(x^{\#} - \hat{x})_{\overline{S}}\|_2 = \inf_{z \in M_k} \|(x^{\#} - \hat{x}) - z\|_2$. Then, using Lemma 16 and (17), we see that

$$\|x^{\#} - \hat{x}\|_2 \leq \|(x^{\#} - \hat{x})_{\overline{S}}\|_2 + \|(x^{\#} - \hat{x})_S\|_2$$

$$\leq \frac{1}{\sqrt{k}}\|(x^{\#} - \hat{x})\|_1 + \frac{\rho}{\sqrt{k}}\|(x^{\#} - \hat{x})_{\overline{S}}\|_1 + \tau\|F(x^{\#} - x)\|_2$$

$$\leq \frac{1+\rho}{\sqrt{k}}\|(x^{\#} - \hat{x})\|_1 + 2\tau\eta$$

$$\leq \frac{(1+\rho)^2}{\sqrt{k}(1-\rho)}\left(2\|\hat{x}_{t(k)}\|_1\right) + \frac{4\tau(1+\rho)}{(1-\rho)}\eta + 2\tau\eta$$

$$= \frac{(1+\rho)^2}{\sqrt{k}(1-\rho)}\left(2\|\hat{x}_{t(k)}\|_1\right) + \left(\frac{4\tau(1+\rho)}{(1-\rho)} + 2\tau\right)\eta$$

Recalling $\tau = 1$ and letting $\rho \to 0$ gives the desired result. ☐

### A.4 DANTZIG SELECTOR

Next we introduce the Dantzig Selector algorithm with an additional constraint. We first prove its recovery guarantees for $\ell_\infty$-norm and then explain the reasoning behind the additional constraint.

***Proof of Theorem 4.*** The proof follows the same structure as the proof of Theorem 3. Therefore we provide a sketch and leave out the complete derivation. Let $0 < \rho < 1$ be arbitrary. Since $F$ is a unitary matrix, for any $S \subseteq [n]$ and $v \in \mathbb{C}^n$, we have

$$\|v_S\|_2 \leq \frac{\rho}{\sqrt{k}}\|v_{\overline{S}}\|_1 + \|v_S\|_2 \leq \frac{\rho}{\sqrt{k}}\|v_{\overline{S}}\|_1 + \sqrt{k}\|v\|_\infty = \frac{\rho}{\sqrt{k}}\|v_{\overline{S}}\|_1 + \sqrt{k}\|F^*Fv\|_\infty$$

The rest of the argument is the same as in the proof of Theorem 3. ☐

