# OpenReview forum: "Compressive Recovery Defense: A Defense Framework for $\ell_0, \ell_2$ and $\ell_\infty$ norm attacks."
_ICLR.cc/2020/Conference — Reject_

### Official Review · AnonReviewer3 · 2019-10-20
**Official Blind Review #3**

**Rating:** 3

**Review:**

The paper studies the problem of the robustness of the neural network-based classification models under adversarial attacks. The paper improves upon the known framework on defending against l_0, l_2 norm attackers.

The main idea of the algorithm is to use the "compress sensing" framework to preprocess the image: Using F, the discrete Fourier transformation matrix, and the algorithm tries to reproduce on every given input x, a vector y with the smallest number of non-zero coordinate such that Fy approximates x. The main algorithms proposed in this paper are sparse iterative hard thresholding (IHT) or base pursuit (BP) which are all quite simple and standardized.

The intuition of the approach is that l_0, l_2 attackers on the original input x can not allude the sparse vector y by too much, thus the recovered vector Fy could have better robustness property comparing to the original input x.


The main concern for me is the experiment in this paper. The author does not provide enough details about how the attacker is trained in their task. It seems that the authors only use the attacker trained on a standard neural network. However, since the authors have a preprocessing algorithm (IHT, BP) on top of the given input, the attacker should in principle tries to attack this pre-processing process as well. Since the pre-processing process is not differentiable, it is, therefore, unclear to me how to define the true robustness of the approach of the authors.

An analog of my argument is if we create an artificial network that has a pre-processing layer that zeros out most of the input pixel, however, if we train an attacker without this knowledge (so it tries to attack a network without this pre-processing), the l_2, l_0 attacker might not be very good for the true network.


After Rebuttal: I have read the authors' responses and acknowledge the sensibility of the statement. However, I still think the algorithm in this paper is merely a "clever" version of gradient masking, which does not give the neural networks real robustness, it is just harder to design attacks on all these discrete operations.



**Experience Assessment:**

I have published in this field for several years.

**Review Assessment: Checking Correctness Of Derivations And Theory:**

I assessed the sensibility of the derivations and theory.

**Review Assessment: Checking Correctness Of Experiments:**

I assessed the sensibility of the experiments.

**Review Assessment: Thoroughness In Paper Reading:**

I read the paper at least twice and used my best judgement in assessing the paper.

---

> ### Author Response · Authors · 2019-11-06
> **Response to Review 3**
>
> Dear Reviewer,
>
> Thank you for the comments and questions. We appreciate the feedback you have provided.
>
> We would like to get clarity on your main concern before we attempt to address it. Could you please clarify what you mean by:
>
> "However, since the authors have a preprocessing algorithm (IHT, BP) on top of the given input, the attacker should in principle tries to attack this pre-processing process as well."
>
> More specifically, is your concern that we have not performed a reverse engineering attack? That is, we have not created an attack that takes CRD (Compressive Recovery Defense) into account before creating an adversarial input?
>
> If this is indeed the question, then we would like to note that we expect creating such a reverse engineering attack to be quite difficult.
>
> We explain why by making some observations first:
>
> 1. IHT can use random initialization which will make the recovery non-deterministic. Therefore, a reverse engineering attack that uses standard backpropagation based approaches may not be effective.
> 2. BP and DS can be cast as semi-definite programming problems. If solved with interior point methods, one can use random initialization of the central path parameter and/or add randomness to the stopping criterion. This will make the recovery non-deterministic once again. Hence, standard attack methods may not be effective.
>
> Given the above, and the fact that we do not have domain expertise on creating attacks, we did not think it would be feasible for us to create a reverse engineering attack against CRD in the present paper. We feel that tackling this interesting problem is better left to a separate paper.
>
>
> We will wait to get more clarity on your concern before providing further feedback.

---

### Official Review · AnonReviewer1 · 2019-10-23
**Official Blind Review #1**

**Rating:** 3

**Review:**

This paper extends the compressive sensing framework introduced in Bafna et al. to handle l1 and l2 attacks. The authors provide theoretical analysis for several recovery algorithms (IHT, BP, DS) and provide experimental result on CIFAR-10, MNIST and Fashion-MNIST.

My major concern is how significant the provided results are. It is indeed interesting to extend the compressive sensing framework to handle l1 and l2 attacks. However, the proposed recovery algorithms are all classical ones, and it is unclear how novel the analysis is, since the authors do no discuss the technical challenges they overcome or the difference between their proof techniques and the previous ones. Also, it would be nice if the authors could discuss the theoretical results in more detail, e.g., how to interpret them and new insights it brings to us.

Moreover, some experimental details are missing. In last paragraph in Section 3.1, the authors say ``We then use both x and x′ to train the network''. How do you do so? Just add both x and x' to the training set?

**Experience Assessment:**

I have read many papers in this area.

**Review Assessment: Checking Correctness Of Derivations And Theory:**

I assessed the sensibility of the derivations and theory.

**Review Assessment: Checking Correctness Of Experiments:**

I assessed the sensibility of the experiments.

**Review Assessment: Thoroughness In Paper Reading:**

I read the paper at least twice and used my best judgement in assessing the paper.

---

> ### Author Response · Authors · 2019-11-06
> **Response to Review 1: Part 1**
>
> Dear Reviewer,
>
> Thank you for the comments and questions. We appreciate the feedback and have tried to address each concern.  Please note that our response is split into two comments.
>
> Reviewer wrote: “Moreover, some experimental details are missing. In last paragraph in Section 3.1, the authors say ``We then use both x and x′ to train the network''. How do you do so? Just add both x and x' to the training set?”
>
> Yes, we simply include both x and x’ in the training set and then train the neural network using stochastic gradient descent. We will state this more clearly in the revised version.
>
> Reviewer wrote: “This paper extends the compressive sensing framework introduced in Bafna et al. to handle l1 and l2 attacks.”
>
> Actually, our extension handles l_2 and l_{\infty} attacks. We have stated nothing about l_1 attacks.
>
> Reviewer wrote: “However, the proposed recovery algorithms are all classical ones, and it is unclear how novel the analysis is, since the authors do not discuss the technical challenges they overcome or the difference between their proof techniques and the previous ones.”
>
> First, we note that we have deliberately used classic recovery algorithms as they are more easily accessible (only for Dantzig Selector have we introduced a simple novel modification). Moreover, since our aim was to provide guarantees against l_0, l_2, and l_{\infty} attacks, we needed recovery algorithms that could be analyzed theoretically for each type of attack.
>
> Algorithm 1 ((k,t)-Sparse Iterative Hard Thresholding): This algorithm was introduced in Baraniuk et al [2] and used by Bafna et al [1] for the compressive defense framework.
>
> Algorithm 3 (Dantzig Selector with additional constraint):
> This is the only algorithm that we have modified in a novel way. We provided an explanation of the additional constraint (paragraph 3 on page 3). This is indeed the only novel algorithmic contribution that we make in this paper.
>
> The main contributions of this work are not in providing a novel recovery algorithms. Instead the main contributions are:
> (a) proving recovery guarantees for the compressive recovery defense (CRD) with existing recovery algorithms
> (b) showing experimentally that the defense does indeed improve performance.
>
> We will make this more clear in the revised version of the paper.
>
> Below we compare our theoretical results and proof techniques to [1], explain why they are non-trivial, and point out some interesting observations. Then we comment on the proofs of Theorems 2, 3, and 4.  We will add this information to the revised version of the paper.
>
> Summary of [1] :
> The authors of [1] provide a defense against against l_0 attacks only. They use IHT to perform recovery with recovery guarantees stated in Theorem 2.2 of [1].  The recovery guarantees provided rely on Theorem 4 of [2]. More specifically, the authors  of [1] show that the matrix used in IHT satisfies the RIP property with \delta <= 0.1. This satisfies the constraints of Theorem 4 in [2] and hence the authors of [1] are able to use the results of Theorem 4 in [2] to get results of Theorem 2.2 of [1].
>
> Comparison to [1]:
> 1. Our results cover l_0, l_2, and l_{\infty} attacks.
> 2. For l_0 attacks, we provide recovery guarantees for IHT (Theorem 1) as well as BP (Theorem 2).
> 3. We provide recovery guarantees for BP as they hold for larger values of k (coefficients to recover) and t (adversarial noise budget). For instance, in the case of MNIST and Fashion-MNIST, IHT (equation (4) of Theorem 1) allows us to set k = 4 and  t = 3, whereas BP (equation (7) of Theorem 2) allows us to set k = 8 and t = 8. We note that even the results of Theorem 1 hold for values of k,t greater than or equal to Theorem 2.2. of [1].
> 4. Since [1] only covers l_0 attacks with IHT, we improve upon the results of [1] in two ways:
>
> i) Equations (3) and (4) of Theorem 1 allow larger values of k and t than Theorem 2.2 of [1]. This is because the authors of [1] use Theorem 4 of [2] to prove their results and this theorem is more restrictive on the values of k and t. We do not rely on Theorem 4 of [2], instead we prove the RIP property using a different proof technique (Theorem 7) and then use Theorem 6.18 of [3] to get equations (3) and (4) of Theorem 1.
>
> ii) Equations (5) and (6) of Theorem 1 provide guarantees for larger values of k and t than equations (3) and (4) by incurring a penalty term in the error bound. To achieve this, we first prove Lemma 9 and then use the result of Lemma 9 to get (5) and (6). The proof of Lemma 9 as well as its application to get (5) and (6) require a novel approach. This can be seen in the proof of Theorem 1.

---

> ### Author Response · Authors · 2019-11-06
> **Response to Review 1: Part 2**
>
> Interesting observations:
> 1. In order to prove the RIP property (Theorem 7), we bound the eigenvalues of the matrix by using normality and the Gersgorin disc theorem. The authors of [1] use direct algebraic manipulation (i.e. triangle inequality and AM-GM inequality) to prove the RIP property (Lemma 3.6 in [1]). In fact, if one replaces certain constants in Lemma 3.6 of [1], the RIP property results  of Theorem 7 and Lemma 3.6 of [1] are identical.
> 2. Equations (5) and (6) of Theorem 1 show that in order to provide guarantees for larger values of k and t, one must incur a penalty. Based on our efforts, we could not find a technique that allowed us to provide guarantees for larger values of k and t without incurring some penalty.
>
>
> Theorem 2:
> Theorem 2 provides guarantees for larger values of k and t. In order to achieve this, we rely on Theorem 4.33 of [3]  (Theorem 12 in our paper) and don’t use a RIP based argument. Instead, we show that the matrix in question does indeed satisfy the hypotheses of Theorem 12 and hence allows to get the bounds of Theorem 2. An examination of the proof of Theorem 2 may reveal it is not entirely trivial.
>
> Theorem 3 and Theorem 4:
> These theorems are based on standard arguments that rely on the robust null space property of the matrix. We first prove the required robust null space property and then follow the classical argument. The general argument used here has indeed been used commonly in compressive sensing literature.
>
>
> Reviewer Wrote: “Also, it would be nice if the authors could discuss the theoretical results in more detail, e.g., how to interpret them and new insights it brings to us.”
>
> We will include such a discussion in the revised version of the paper.
>
>
>
>
> [1] Mitali Bafna, Jack Murtagh, and Nikhil Vyas. Thwarting adversarial examples: An l 0-robust sparse fourier transform. In Advances in Neural Information Processing Systems, pp. 10075–10085, 2018.
>
> [2] Richard G. Baraniuk, Volkan Cevher, Marco F. Duarte, and Chinmay Hegde. Model based compressive sensing. IEEE Trans. Information Theory, 56(4):1982–2001, 2010.
>
> [3] Simon Foucart and Holger Rauhut. A Mathematical Introduction to Compressive Sensing. 2017.

---

### Official Review · AnonReviewer2 · 2019-10-24
**Official Blind Review #2**

**Rating:** 6

**Review:**

This paper extends the compressive recovery defense framework introduced by Bafna et al. (2018), which is mainly against l_0 attacks, to l_2 and l_∞ attacks. They provide guarantees for some recovery algorithms in the case of different kinds of norm bounded noises. The difference between their work and the previous work is clearly clarified.

Overall, this paper is a follow-up work towards Bafna et al. (2018) but with better theoretical guarantees and ample experiment results to support their robustness against various popular attacks. Given their contribution and inspiration for future work, I think this paper could be accepted to the 2020 ICLR conference.

In Section 3.2 Recovery Algorithms, the author clearly states three algorithms including IHT, BP, DS, and their modification from the standard ones, but fails to compare the differences between these algorithms. It is not clear about the author’s motivations to proposes these different recovery algorithms and whether their performance varies from each other also remains unknown. Maybe some analysis about their disadvantages and advantages in varied conditions of attacks could be necessary.

In the Section 3.4 Comparison to Related Work, the author mentions many works aiming at defending against adversarial inputs. However, Bafna et al. (2018) is the only work here that has something to do with compressive sensing. I think maybe the paper should involve some related work here regarding theory of compressive coding besides Bafna et al. (2018). And how they are combined to the defense against adversarial inputs. It would help the readers to have better understanding towards the novelty and breakthroughs in this aspect.

For the experiments, it would be better to have the comparisons between the proposed algorithm and related methods. Also, the proposed IHT and DS are modified versions. What are the differences in experiments?

Minor comments:
- Page 2: the line above the equation 1 ‘meaning that x_t (k)…..’, it could be (x_t ) ̂(k)
- Page 6: in the explanation of figure 2, the adversarial inputs are second column and fifth column not fourth column.


**Experience Assessment:**

I have read many papers in this area.

**Review Assessment: Checking Correctness Of Derivations And Theory:**

I assessed the sensibility of the derivations and theory.

**Review Assessment: Checking Correctness Of Experiments:**

I assessed the sensibility of the experiments.

**Review Assessment: Thoroughness In Paper Reading:**

I read the paper at least twice and used my best judgement in assessing the paper.

---

> ### Author Response · Authors · 2019-11-06
> **Response to Review 2**
>
> Dear Reviewer,
>
> Thank you for the comments and questions. We appreciate the feedback you have provided. We have attempted to address your concerns below and also ask for clarifications as required.
>
> Reviewer wrote: “In Section 3.2 Recovery Algorithms, the author clearly ... could be necessary. “
>
> Since our aim was to provide guarantees against l_0, l_2, and l_{\infty} attacks, we needed recovery algorithms that could be analyzed theoretically for each type of attack. The algorithm we choose depends on the norm used to bound the attack vector. We explain in detail below.
>
> l_0 attacks:
> Once the problem has been reformulated as in equation (2), the vector that we need to recover is (k,t) sparse. This is only possible since the attack vector is t-sparse, that is it is bounded in l_0 norm. Therefore, IHT works due to the sparsity of the attack vector which is only guaranteed in the l_0 attack case. In particular, we cannot use IHT for the l_2 or l_{\infty} attack case as the recovery guarantees would be poor (if derivable at all).
>
> We also provide recovery guarantees for BP as the results of Theorem 2 hold for larger values of k and t. This acts as an improvement over equation (3) and (4) in Theorem 1.
>
> l_2 attacks:
> Noting that l_2 norm behaves very nicely with unitary matrices (acts as an isometry) we use BP for l_2 norm attacks as it allows us to leverage the fact that the matrix F is unitary. Using this and the fact that constraints in BP are designed to handle l_2 norm noise, we are able to get the required recovery guarantees.
>
> l_{\infty} attacks:
> First we note that both BP and DS can be posed as semi-definite programming problems. However, the constraints of DS are designed to handle l_{\infty} norm noise while those in BP are designed to handle l_2 norm noise. Therefore, we use DS when the attacker is only bounded in l_{\infty} norm.
>
> Essentially, each recovery algorithm treats the attack vector as noise and each algorithm is best suited to handle noise bounded in a specific norm (l_0, l_2, or l_{\infty}).
>
> In general the time complexity of IHT is better than the time complexity of BP and DS as the latter are semi-definite programming problems and require expensive matrix operations. However, if required, we can provide an analysis on the time and space complexity of each.
>
>
> Reviewer wrote “I think maybe the paper should involve some related work here regarding theory of compressive coding besides Bafna et al. (2018). And how they are combined to the defense against adversarial inputs. It would help the readers to have better understanding towards the novelty and breakthroughs in this aspect.”
>
> Yes, we agree and will definitely add more related work on compressive sensing. We were apprehensive in doing so earlier due to space constraints.
>
> Could you please clarify what you mean by the below?
>
> For the experiments, it would be better to have the comparisons between the proposed algorithm and related methods.
>
> If your concern is that we should compare our results to those of other defense methods, then we can definitely do so. We can add a comparison in the next revised version of the paper.
>
>
> Reviewer wrote “Also, the proposed IHT and DS are modified versions. What are the differences in experiments?”
>
> We have only modified the DS algorithm and use the same IHT as used by Bafna et al (and introduced by Baraniuk et al). For DS, we have added the additional constraint that improves reconstruction quality. We can definitely run an experiment that visually (or in a norm) shows the improvement in reconstruction with the added constraint.
>
> Thank you for pointing out the minor comments, we will address both points in the revised version of the paper.
>
>
> [1] Mitali Bafna, Jack Murtagh, and Nikhil Vyas. Thwarting adversarial examples: An l 0-robust sparse fourier transform. In Advances in Neural Information Processing Systems, pp. 10075–10085, 2018.
>
> [2] Richard G. Baraniuk, Volkan Cevher, Marco F. Duarte, and Chinmay Hegde. Model based compressive sensing. IEEE Trans. Information Theory, 56(4):1982–2001, 2010.

---

### Author Response · Authors · 2019-11-13
**Revision Uploaded**

Dear Reviewers,

We have uploaded a revision with the following changes:

Abstract:
-Clarified that we use a modified version of DS.

Section 2:
-Specified that we use standard BP, (k,t)-sparse IHT, and DS with an additional constraint.
-Minor changes to structure.

Section 3.1:
-Fixed typo pointed out by Reviewer 2
-Updated last paragraph to address Reviewer 1's concern on training procedure.

Section 3.2:
- Added an explanation justifying the use of each algorithm.
- Noted that the experiment section would illustrate why DS with additional constraint performs better.
- Updated paragraph on reverse engineering attacks to account for Reviewer 3's concerns.

Section 3.3
- Added discussions on how to interpret the results for each theorem to address Reviewer 1's concerns. They are provided after the statement of each theorem.
- Added comparison of  (3) and (4) to Theorem 2.2 of Bafna et al. (2018), discussed proof techniques for (5) and (6), and  provided motivation for Theorem 2. This was also done to address Reviewer 1's concerns.

Section 3.4
-Added paragraph on related works in compressive sensing to address Reviewer 2's concern.

Section 4.1.2:
-Fixed typo in Figure 2 description as pointed out by Reviewer 2.

Section 4.3:
- Updated first and second paragraphs to account for additional constraint.
- Added Figure 4 to illustrate the benefit of additional constraint as suggested by Reviewer 2.
- Updated Table 4 to illustrate the benefit of additional constraint as suggested by Reviewer 2.

---

### Decision · Program_Chairs · 2019-12-19

**Decision:**

Reject

**Comment:**

After reading the author's response, all the reviwers still think that this paper is a simple extension of gradient masking, and can not provide the robustness in neural networks.